# Neurophysiologic Reactions during Heart Rate Variability Biofeedback Session in Adolescents with Different Risk of Internet Addiction

**DOI:** 10.3390/ijerph19052759

**Published:** 2022-02-27

**Authors:** Denis Demin, Liliya Poskotinova

**Affiliations:** N. Laverov Federal Center for Integrated Arctic Research of the Ural Branch of the Russian Academy of Sciences, 163069 Arkhangelsk, Russia; denisdemin@mail.ru

**Keywords:** Internet addiction, electroencephalogram, heart rate variability biofeedback, adolescents

## Abstract

The aim of this study was to determine electroencephalogram (EEG) in a session of heart rate variability biofeedback (HRV BF) in adolescents with different Internet addiction (IA) risks. In total, 100 healthy adolescents aged 16–17 years with minimal risk of IA (Group I, 35%), pronounced risk of IA (Group II, 51%), and stable pattern of IA (Group III, 14%) using the Chen Internet Addiction Scale were examined. HRV and EEG parameters were determined at baseline (5 min), and then during the short-term HRV BF session (5 min), in order to increase the total power (TP, ms^2^) of the HRV spectrum. Against the background of an increase in the TP and a decrease in sympathetic activity, an increase in alpha EEG was revealed, especially in Groups I and II. The greatest increase in the power of beta_1_-activity of EEG in the frontal, central, and temporal brain regionswas found in Groups I and II. In adolescents with a pronounced risk of IA, HRV BF is accompanied by a severe activation of the brain systems, while in persons with a stable type of IA, the least brain reactivity is shown, especially in the beta_1_ EEG band.

## 1. Introduction

In today’s world, ubiquitous access to a variety of information is achieved through the use of Internet resources, but their excessive use can lead to Internet addiction (IA). Data on the prevalence of IA among adolescents and young people differ depending on the region of the world, from 2.6% in Europe and North America [1,2,3] to 10.9% in the Middle East [2] and 21.2% in South Asia [4]. According to Russian researchers, the prevalence of IA among adolescents reaches 22.6% [5]. The long-term consequences of IA in adolescents manifest in their significant social maladjustment: difficulties in acquiring a profession, work, failures in interpersonal relationships, and an inability to create a family. The neurophysiologic mechanisms by which excessive use of the Internet can affect the structure, function, and cognitive development of the brain are currently not fully understood. In recent studies, the importance of neurobiological factors in the formation of IA has been noted [6,7]. Changes in various brain structures identified by fMRI in adults with IA have been identified [8]. The values of various neuropsychological disorders among adolescents IA were also noted [5].

Most of the IA psychological correction programs offered by researchers [1,3] are built to predominantly take into account psychological but not biological predictors of Internet addiction, which makes it difficult to identify the main human target organs and systems for the correction of this condition. One such targeted human system [9] is the autonomic nervous system (ANS). Evidence suggests that there is a predominance of activation of the sympathetic nervous system in people with excessive use of Internet resources, and the overall heart rate variability (HRV) is lower in schoolchildren with IA than in their peers without IA [10]. In this regard, the technologies for activating the parasympathetic division of the ANS ensure the optimisation of cardiopulmonary interactions and allow the autonomic tone to be optimised. One of the promising methods of non-drug correction is heart rate variability biofeedback (HRV BF) technology, in which there is an increase in vagal influences on the heart rate and a decrease in the phenomena of sympathicotonia and emotional stress [11]. The severity and reactivity of the main rhythms of the electroencephalogram (EEG) may reflect the nature of the functioning of thalamic–cortical and neural–visceral connections in the implementation of HRV BF training. Due to the pronounced behavioural regulation of respiration under control of the volitional sphere [12], the ability for effective self-regulation with biofeedback in healthy people through breathing is possible even with a single session of biofeedback, against the background of which significant changes in brain activity occur. At the same time, short-term HRV BF training to increase the total power of the HRV spectrum can be considered a kind of cognitive test, when a person is able to implement a mental strategy to purposefully change his visceral functions in a short period of time [13].

Despite the fact that the total power of the heart rate variability spectrum reflects the totality of all mechanisms of autonomic regulation of the heart rate, this indicator primarily reflects the contribution of respiratory waves and vagal regulation with short recordings (up to 5 min), which is a physiological analogue of the indicator, and the standard deviation of heart inter-beat intervals [14]. Earlier, in a controlled study, we showed that more pronounced changes in the brain bioelectric activity occur after a single session of HRV BF than that in individuals who were in a resting state without HRV BF, particularly in normalisation of brain activity and improvements of cortical stability in the form of an increase in alpha EEG activity [15]. Optimisation of EEG parameters after HRV BF training in adolescents with basic sympathicotonia revealed that this is especially important both after a single HRV BF session and after a course of 10 HRV BF sessions [16].

It is known that the formation of any type of addictive behaviour is based on changes in the functioning of the brain [17]; therefore, the identification of the most sensitive brain regions to the process of HRV BF training in individuals with different risks of developing IA is an urgent task. This will further allow the potentially adverse consequences for the brain–heart system of excessive use of the Internet to be minimised. However, the process of self-regulation using biofeedback technologies can have similar physiological effects on cardiac activity but have different effects on bioelectrical brain activity. This may be due to both different variants of the initial autonomic nervous tone and various changes in neurotransmitter activity due to excessive use of Internet resources by adolescents. The scientific hypothesis of this study was that the functioning of the ‘brain–heart’ system during the implementation of the biofeedback process will be different in individuals with Internet addiction and those with minimal risk of developing Internet addiction. The aim of the present study was to investigate EEG changes after a single HRV BF session in order to increase the general heart rate variability in adolescents with different risks of developing IA.

## 2. Materials and Methods

All participants aged 16–17 in this non-randomised study were natives of cities of the Russian Federation and were students (*n* = 100, Europoid residents) of general education schools (2020–2021). This study is a follow-up to our previous study where similar selection criteria were used [18,19]. This group was initially formed as a group of healthy individuals according to medical criteria based on the opinion of the paediatrician, as reflected in the personal medical documentation in the medical office of the school. Due to sex- and age-related differences in the reactivities of the cardiovascular system in adolescents, we decided to form a homogeneous group in terms of sex and age [18,19]. The ratio of boys and girls in different age groups was as follows: 16 yearsold, 21/26; 17 yearsold, 24/29. The number of human samples (*n* = 100) is based on the exclusion of 3 study participants. Missing data were in these 3 participants due to missing data on heart rate variability and EEG.

After the formation of this group, it was found that all subjects had been brought up in families where both parents were present, and the families lived in apartments in multi-storey buildings. Each participant had access to the Internet through both a personal phone and a computer at home. The parents of all the participants were working citizens, and the levels of financial income of the families in which the respondents were brought up corresponded to the average income in the region of residence. There were no students who could not cope with the school curriculum, and none of them studied according to individual programs due to physical or mental health disabilities. None of the participants smoked or consumed alcoholic drinks. The study participants had not committed any offences and did not need police or psychological supervision in connection with the manifestation of aggression, depression, suicidal behaviour, or the use of psychoactive substances [18,19].

Anthropometric parameters (height and body mass index (BMI)) were measured for each participant in the school medical office. BMI was defined as the body mass in kilograms divided by the square of the body height in metres (kg/m^2^). Height, weight, and BMI were measured using certified medical equipment with electronic scales and a height metre (REP+VMEN-200–100-D1-A, Factory Twes, Tambov, Russian) [18,19]. The anthropometric parameters of the examined individuals were above the 97th and below the 3rd percentiles according to the height-for-age and BMI-for-age scales for 5–19-year-old boys of the corresponding age according to the criteria of the World Health Organisation [20].

The total time for using the Internet (for the purpose of learning in the school curriculum and for entertainment) averaged 2.5 h for 27.3% of individuals, 5.5 h for 38.4% of individuals, and 6 or more hours for 34.3% of individuals every day. The entertainment time included social media time, watching video content and news, and online games [18,19]. For the purpose of entertainment, 38.5% of participants used the Internet for more than 3 h a day (up to 5.5 h), 32% used the Internet for 2–3 h a day, and 28.2% used the Internet for up to 2 h a day. 

All participants were administered the Chen Internet Addiction Scale (CIAS) [21] and the Russian version by Feklisov and Malygin [22]. The CIAS comprised a questionnaire with 26 items and a four-point Likert scale, ranging from 1 point (does not match my experience) to 4 points (definitely matches my experience). Thus, the minimum CIAS value was 26, and the maximum value was 104. Respondents with CIAS scores above 64 points were considered to have a stable pattern of IA, scores of 43 to 64 were associated with a moderate risk of developing IA, and those less than 43 points indicated a minimal risk of developing IA [18,19]. The samples formed were equivalent in terms of gender composition and social characteristics.When studying Internet-dependent behaviour in Russian high school students, the reliability measures of the test CIAS (Cronbach’s alpha)ranged from 0.757 to 0.9, depending on the CIAS subscales [22]. In other words, the CIAS test presented a sufficient degree of reliability.

Heart rate variability (HRV) parameters were recorded in the sitting position using the ‘Varicard’ device (Ramena Company, Ryazan, Russia). We determined HR (bpm), total power (TP, ms^2^) of the HRV spectrum, and Baevsky Stress Index (SI, units) [14], reflecting the level of sympathetic effects on the heart rhythm. SI was calculated by the formula SI = Amo50/2×VAR×Mo, where Mo(Ms) is the cardiointerval value dividing the cardio-interval-gram series in half, VAR is the variation range between the minimum and maximum values in the cardio-interval-gram series, and Amo50 (%) is the amplitude of mode (the most frequent R–R intervals. HRV parameters were recorded in rest (5 min) and during the HRV biofeedback (HRV BF) once a session (5 min). All participants had a 5-minute preliminary HRV BF training session. During the HRV BF session, the participants had to maintain a state of calmness and muscle relaxation and breathe with a deep calm inhalation and a smooth, slow exhalation. The subjects performed visual control of TP in the form of a number and graphical trend on a computer monitor, which should have increased with effective HRV BF sessions [23].

EEG was recorded during the final 2 min of each stage of the study, in a sitting position, in a state of calm wakefulness with closed eyes using a Neuron-Spectrum-4/VPM electroencephalograph (NeuroSoft, Ivanovo, Russia). A monopolar scheme of 16 standard leads according to the international system 10–20 with ear reference electrodes was used. When evaluating the EEG, artifact-free 60-second recording segments were identified at each stage of the study. The EEG spectrum was analysed for theta (4.0–7.5 Hz), alpha (8.0–13.5 Hz), and beta_1_ (14.0–24.0 Hz) bands. The quantitative assessment of the EEG spectrum in each frequency band was carried out according to the values of the absolute spectral power (μV^2^) in the frontal (F3 F4), central (C3 C4), temporal (T3 T4), and occipital cerebral regions (O1 O2).

Statistical data wereprocessed using STATISTICA software (StatSoft Inc., Tulsa, OK, USA, v. 13.0). The description of quantitative HRV parameters was carried out with an indication of the median and range of values corresponding to the 25th and 75th percentiles (lower and upper quartiles). Quantitative variables in the independent groups were compared using the Kruskal–Wallis test (while comparing three groups, *p* < 0.017). HRV parameters in each group between HRV BF value and baseline value were compared using the Wilcoxon test (*p* < 0.05). EEG data are presented as a percentage (% increase in the EEG power HRV BF value in relation to the baseline value), but the significance levels on the graph are presented between the absolute values of the EEG spectrum power at HRV BF session and these values at rest in each group using the Wilcoxon test (*p* < 0.05), as well as between groups I and II during the HRV BF session using Mann–Whitney U test (*p* < 0.05).

## 3. Results

The total sample of adolescents surveyed was divided into three groups in accordance with the number of points they received for the questionnaire survey using the CIAS scale. Group I, with minimal risk of developing IA, included 35 adolescents, for whom the CIAS was 37.0 (32.0; 40.0), Group II, with a tendency to develop IA, included 51 adolescents (50.0 (46.3; 55.8) points), and Group III, with formed IA, was composed of 14 adolescents (69.0 (67.3; 72.0) points) (*p* < 0.001).

The baseline HRV indices did not statistically differ in the groups; however, the SI value in individuals with signs of IA (Group III) was slightly higher than that in individuals from other groups. In 45% of individuals in Group III, the SI was more than 150 units, which can be regarded as a state of sympathicotonia. During the HRV BF session in adolescents from all groups, the values of the controlled indicator (TP) significantly increased in comparison with the background values (*p* < 0.001) (Table 1). 

SI also significantly decreased in all adolescents (*p* < 0.001). Heart rate did not change significantly in representatives of all groups. 

An increase in the spectral power of the studied frequency band EEG during the HRV BF session in comparison with the baseline values in adolescents from all groups was observed (Figure 1).

In Group I participants, the power of theta EEG activity changed insignificantly (*p* > 0.05). In Group I participants, the power of EEG theta activity changed insignificantly, up to 5% (*p* > 0.05). In adolescents from Group II, with an increase of 10–15%, EEG theta activity increases in all brain regions in comparison with the background values (*p* = 0.032–0.001); it was most pronounced in the frontal (F3), temporal (T3) and central (C3) regions on the left (*p* < 0.001). A higher value of EEG power in Group II, compared with Group I, was noted in the left temporal (T3) and right central (C4) regions (*p* = 0.015). In adolescents from Group III, a significant increase in the EEG of theta activity was found in the left frontal (F3, *p* = 0.032), right temporal (T4, *p* = 0.025), and occipital cerebral regions (O1 O2, *p* = 0.018). Significant increases in alpha-band EEG power during the HRV BF session occurred in participants from all groups. The greatest increases in values (18–30% relative to background values) were noted in adolescents of Group I bilaterally in the frontal (F3 F4), central (C3 C4), and temporal (T3 T4) brain regions (*p* < 0.001), as well as in the central regions (C3 C4), and were significantly higher than in Group II (*p* = 0.014). In adolescents from Group II, a significant increase in alpha activity was found in the frontal, central, and left temporal regions (*p* = 0.040–0.001). In adolescents from Group III, the percentage increases in EEG alpha activity were slightly less (10–20%) and significantly increased in HRV BF only in the left frontal (F3, *p* = 0.028) and temporal (P3, *p* = 0.035) regions. Topical EEG features in which increases in the spectral power of the EEG alpha band in adolescents of all groups under consideration occurred without significant changes in the occipital regions were noted. The percentage increases in the power of beta_1_ oscillations of the EEG in adolescents of Groups I and II were more pronounced (up to 26%), and in adolescents of Group III, they were minimal (7–20%). At the same time, in adolescents of Group I, the beta_1_ activity of the EEG significantly increased in the frontal, central (*p* < 0.001) and left temporal (*p* = 0.009) regions; in adolescents of Group II, in the frontal, central, temporal (*p* < 0.001) and left occipital (*p* = 0.008) cerebral regions; in adolescents of Group III, only in the left temporal (*p* = 0.004) cerebral region.

## 4. Discussion

The results indicate that all adolescents were able to successfully complete a single HRV BF session and increase the reserve of the parasympathetic autonomic nervous system. At the same time, the participants achieved a state of general relaxation, calmness, mental comfort, and emotional balance. It is important to note that the initially increased sympathetic activity in individuals with the Internet addiction pattern (Group III) also significantly decreased, and SI became comparable with this indicator in individuals from other groups. During short-term recordings (up to 5 min), the TP parameter contains the minimum contribution of non-periodic (non-respiratory) waves. Therefore, it can be assumed that the effect of HRV BF is achieved by increasing the contribution, first of all, of the respiratory and baroreflex components of the HRV spectrum [14]. Other authors have shown that relaxation and meditation accompanying biofeedback training can have pronounced effects on both ANS and brain activity [11,24,25]. The absence of significant changes in EEG theta activity in adolescents without IA (Group I) may indicate the resistance of their subcortical diencephalic brain structures to changes in the ANS balance during the HRV BF procedure. In adolescents at risk of developing IA (Group II) and, to a lesser extent, with a stable IA pattern (Group III), there is a pronounced activation of the limbic–thalamic, hippocampal–cortical systems (increased EEG theta activity) under the given conditions of the HRV BF procedure. The combined enhanced influence of thalamic and brainstem structures on the bioelectric genesis of the cortex results in increased alpha activity in the frontal and central regions of the brain [26]. It is known that, in a person who is in states of meditation and relaxation, an increase in the synchronisation of theta waves and slow alpha1 waves in the frontal regions is most often noted [24]. Moreover, an increase in the power of the alpha_1_ EEG rhythm in the parietal–central cerebral region is accompanied by an increase in the coherence of the heart rate during meditation [25]. Neuroimaging technologies have also revealed functional connections between the frontal and central cerebral regions: the medial prefrontal cortex, anterior cingulate gyrus, amygdala, and hippocampus with autonomic changes in the cardiovascular system [27]. At the same time, activation of the left hemisphere is associated with an increase in the activity of the parasympathetic division of the ANS [28] and with positive emotions [29]. In general, an increase in EEG alpha activity indicates optimisation of the cortical–subcortical relationship, contributing to a decrease in the activity of the sympathetic division of the ANS in HRV BF. In any addictive disorder, a person is in a state of mental discomfort most of the time, and the process of satisfying addiction usually improves his mental state and leads to an increase in the alpha and theta EEG activity; the reason for this is an increase in the synthesis of endorphins [17]. At the heart of addiction to social networks and Internet games is the activation of the ‘dopamine feedback loop’, which is formed when the feedback system is closed on itself. To maintain this, developers of Internet resources use bonuses, winnings, incentives, likes, etc., which cause pleasure and addiction in humans [3]. Such a ‘loop’ in some cases is called compulsive and/or obsessive. It includes the cerebral cortex, basal ganglia, thalamus, epithalamus, and midbrain, through which dopaminergic activity alters human motivation [30]. The sleep deficit seen in IA has an additional negative effect on the prefrontal cortex, as a result of which the addict cannot think rationally and strategically [5]. An increase in EEG activity during HRV BF may reflect the process of reducing internal anxiety, a decrease in the subconscious need for addictive behaviour, and an increase in psychological comfort. The available data suggest that the biofeedback effect is achieved through the stimulation of the same endorphin secretion and the reward system [17,30], and HRV BF in this case is a model of effective behaviour, as opposed to addictive behaviour. HRV BF can further increase the activity of the prefrontal cortex by distinguishing between false rewards (which make the adolescent become frustrated and addicted) and true rewards (which give them life value).

A significant increase in beta_1_ activity EEG in the frontal, central, and temporal brain regions in adolescents of Groups I and II indicates the wide involvement of the sensorimotor cortex, media-basal, and emotiogenic brain structures [31] in the implementation of an individual strategy for the effective biofeedback as a type of cognitive activity. Mechanisms of EEG beta oscillations can be determined by the properties of neurons in both the cortex and the thalamus, as well as intracortical and thalamocortical interactions; in addition, there are data on the generation of a beta_1_ rhythm in the 15–20 Hz band in the CA1 region of the hippocampus [32]. Previously, we observed the absence of significant reactivity of high-frequency EEG in response to the HRV BF procedure in adolescents with symptoms of sympathicotonia [33]. In the presented study, among individuals with a stable pattern of IA (Group III), there were more participants with sympathicotonia in terms of SI, which could also be associated with the low reactivity of brain structures in the high-frequency EEG range. This may be due to both a decrease in the emotional significance of the HRV BF procedure and a possible neurotransmitter deficiency in individuals with IA. The cerebral mechanisms of cardiovascular control may depend on the position of the human body (lying and standing positions). In our study, in order to assess the dynamics of HRV indicators in the background and with HRV BF, we considered it important to use an intermediate body position in the study participants - the ‘sitting’ position. As a development of the research topic, it seems important in the future to use HRV BF technologies in various body positions to optimise both cerebral haemodynamics and brain bioelectrical activity in individuals with sympathicotonia and excessive use of Internet resources. 

## 5. Conclusions

Thus, in the studied sample of adolescents *(n* = 100), only one-third of the participants had a minimal risk of IA (35%), the largest number of individuals had a pronounced risk of IA (51%), and 14% of people had a stable IA pattern. Performing a single HRV BF session increased overall heart rate variability and decreased sympathetic activity, which is especially important for individuals with IA signs. The greatest stability of the subcortical diencephalic brain structures (stability of EEG theta activity) during HRV BF was shown in adolescents with minimal risk of IA, and the greatest increase in EEG theta activity in the frontal, central, and temporal cerebral regions was observed in the group with a pronounced risk of IA. In the majority of all adolescents, a significant increase in EEG alpha activity was revealed in frontal and central brain parts, which reflects the combined enhanced influence of thalamic and stem structures on the bioelectric genesis of the cortex. In adolescents with minimal and pronounced risk of IA, the implementation of the HRV BF effect occurred against the background of a maximum increase in the power of beta_1_ activity of the EEG with wide involvement of the sensorimotor cortex and media-basal brain structures. In adolescents with an IA pattern, the percentage increase in EEG beta_1_ activity was minimal, which may indicate a relative neurotransmitter deficiency and a decrease in the reactivity of emotiogenic structures during the HRV BF procedure. In general, the search for the leading rhythmogenic cerebral structure is very promising, a change in the state of which will effectively initiate the restructuring of all other cerebral structures during the implementation of HRV BF in adolescents who are intensively involved with the Internet.

## Figures and Tables

**Figure 1 ijerph-19-02759-f001:**
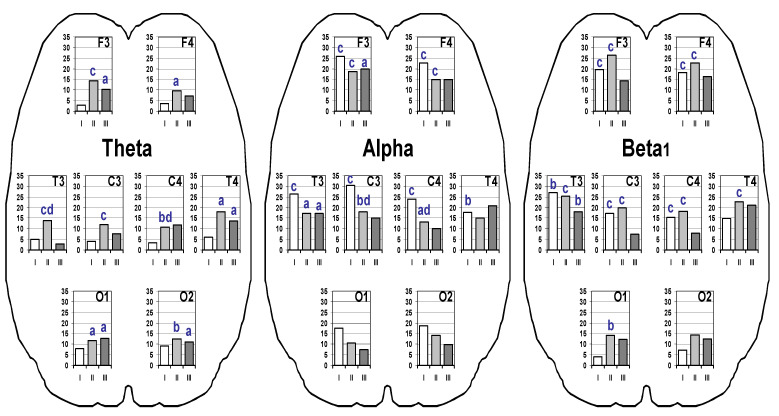
Percentage gains (in %) of the EEG spectral power in frequency EEG bands in the dynamics of HRV BF session in adolescents with different risks of Internet addiction (IA). F3, F4, C3, C4, T3, T4, O1, O2–left and right frontal, central, temporal and occipital EEG points. White bars, group with a minimal IA risk (I); light grey bars, group with a tendency towards IA (II); dark grey bars, group with a stable IA pattern (III). Statistically significant difference between baseline and after HRV BF session parameters: ‘a’, *p* < 0.05; ‘b’, *p* < 0.01; ‘c’, *p* < 0.001; ‘d’, *p* < 0.05 between Groups I and II after the HRV BF session.

**Table 1 ijerph-19-02759-t001:** Changes in heart rate variability parameters in the dynamics of heart rate variability biofeedback (HRV BF) session in adolescents with different IA risks ^a^.

Variables	Group	Baseline	HRV BF
SI, units	I	119.0 (71.9; 208.4)	78.1 (47.0,129.4) ^b^
	II	115.2 (78.7; 193.6)	60.2 (41.3; 101.4) ^b^
	III	183.3 (106.8; 329.1)	83.6 (52.1; 114.0) ^b^
TP, ms^2^	I	1899 (1468; 2923)	83.6 (52.1; 114.0) ^b^
	II	1895 (1350; 2961)	6135 (3865; 8793) ^b^
	III	1565 (1170; 2277)	5078 (3445; 6743) ^b^

SI: Stress Index; TP: total power HRV; HR: heart rate. ^a^ Data are presented as median (lower and upper quartiles). ^b^ Statistically significant difference between baseline and after HRV BF session parameters: *p* < 0.001, Wilcoxon test.

## Data Availability

Not applicable.

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
