# Peer review of "Neurophysiologic Reactions during Heart Rate Variability Biofeedback Session in Adolescents with Different Risk of Internet Addiction"

_ijerph, 2022, doi:10.3390/ijerph19052759_

Round 1

Reviewer 1 Report

1.Lack of theoretical clarity

The authors made a good summary of the literature dealing with IVs, but there are no guiding theory/hypotheses. Simple citing studies that have various findings is not enough.

  1. Sample

Please provide more information regarding sample characteristics.

  1. Reliability and validity

Please provide more information regarding variable used.

Are scales used in this study validated for high school students in original scale development and Russian samples as well?

  1. Other issues.

There is no information about missing data.

Author Response

We would like to thank the esteemed reviewer for critically analyzing the manuscript.

Point 1: Lack of theoretical clarity. The authors made a good summary of the literature dealing with IVs, but there are no guiding theory/hypotheses. Simple citing studies that have various findings is not enough.

Response 1:

The process of self-regulation using biofeedback technologies can have similar physiological effects on cardiac activity, but have different effects on bioelectrical brain activity. This may be due to both different variants of the initial autonomic nervous tone, and various changes in neurotransmitter activity due to excessive use of Internet resources by adolescents. The scientific hypothesis of this study was that the functioning of the “brain-heart” system during the implementation of the biofeedback process will be different in individuals with Internet addiction and those with a minimal risk of developing Internet addiction.

We formulated this hypothesis and included it in the introduction section (this is highlighted in red).

Point 2: Please provide more information regarding sample characteristics.

Response 2: We tried to indicate all the characteristics of human samples at the beginning of the Materials and Methods section, such as: age, ethnicity, gender, anthropometric characteristics, health and habitat characteristics, as well as the time and purpose of using Internet resources.

Point 3: Please provide more information regarding variable used. Are scales used in this study validated for high school students in original scale development and Russian samples as well?

Response 3: CIAS scales used in this study validated for high school students in original scale development and Russian samples as well.

When studying Internet-dependent behavior in Russian high school students, the reliability measures of the test CIAS (Cronbach’s alpha) was ranged from 0.757 to 0.9, depending on the CIAS subscales [20 - http://medpsy.ru/mprj/archiv_global/2011_6_11/nomer/nomer03.php]. In other words, CIAS test is presented with a sufficient degree of reliability.

We have added information about this to the Materials and Methods section (this is highlighted in red).

 Point 4: There is no information about missing data.

Response 4: The number of human samples (n=100) is based on the exclusion of 3 study participants. Missing data were in these 3 participants due to missing data on heart rate variability and EEG.

We have added information about this to the Materials and Methods section (this is highlighted in red).

Reviewer 2 Report

The objective of this article is to examine electroencephalogram (EEG) and heart rate variability biofeedback (HRV BF) in adolescents with different Internet addiction (IA) risks. See abstract lines 9-10. The Chen Internet Addiction Scale (CIAS) in the Russian version that was developed by Malygin and Feklisov, is the main method used to inquiry the topic.

To do their analysis the authors have taken into consideration:

  • the ubiquity of access to a variety of information is achieved through the use of Internet resources,
  • long-term consequences of IA in adolescents,
  • the neurophysiological mechanisms employed with an excessive Internet use may affect functional and cognitive development of the brain,
  • the various neurobiological factors and neuropsychological disorders among adolescents with IA.

See the Introduction, lines 24-37.

Since the predominance of the sympathetic nervous system’s activation has been observed in individuals with Internet overuse, the authors borrow transparently many ideas for this research from Krivonogova, O. et al. (2021), which has focused on HRV (heart rate variability) as “one of the most important indicators of adaptive processes in the human organism.”

In addition, neural oscillations, or brainwaves, are rhythmic or repetitive patterns of neural activity in the central nervous system, may include many still-on-exploration role such as feature binding, information transfer mechanisms and the generation of rhythmic motor output.

Then, the authors address themselves to inquiry in a sample of participants aged 16-17 in 2020-2021 both HRV BF and EEG, to check the functioning of thalamic-cortical and neutral-visceral connections as well as to detect eventual anomalies in adolescents due to IA risks. See lines 38-80.

Thalamocortical oscillation is associated with the appearance of specific mental states depending on the frequency range of the most prominent oscillatory activity, gamma most associated with conscious, selective concentration on tasks, learning (perceptual and associative), and short-term memory.

Sample’s information is collected at Section 2. Materials and Methods. Other descriptive data modelled on Krivonogova, O. et al. (2021) are furnished at lines 91-111.

People in the sample used Internet both through a personal phone and a computer at home, and their parents were all working citizens with an average income in the region of residence.

Average time spent on Internet ranged from about 2.5 hours until 6 or more hours a day, for learning, reading content and news, entertainment included social media time.

The administered CIAS with a four-point Likert scale to respondents served to state whether questions “do not matched respondents’ experience”/”definitely matched…” and scored from 26 to 104 points. Then, the respondents were recorded about their HRV.

Through STATISTICA software the authors have applied non-parametric tests, such as Mann-Whitney U-Wilcoxon, to data with the aim at ranking raw values.

In their Section 3. Results, the authors have divided the total sample into three groups in accordance with the number of points they received for the questionnaire survey using the CIAS scale. See lines 165-170.

Table 1 at line 177 indicates changes in HRV parameters in people with IA risks expressed through the Baevsky Stress Index (SI, units) and the Total Power (TP, ms2) of the HRV spectrum.

Figure 1 at line 190 shows percentage gains (in %) of the EEG spectral power in frequency EEG bands in the dynamics of HRV BF session in adolescents with different IA risks.

The authors give interesting physical condition feedbacks about their three groups at lines 202-228.

In general, it is noticed in these subjects with IA risks a dopaminergic activity alters human motivation, other than cause sleep deficit, depletion of rational and strategic thinking, among other effects, may be moderated through a more relaxed stimulation of the BF effect. See lines 247-281. The authors conclude their paper prizing on the search for the leading rhythmogenic cerebral structure such as the EEG as a combined asset for a stress indicator tool based on HRV. See lines 319-322.

CHANGE REQUEST:

  1. Please add a space after “104.” at line 123,
  2. Please add a space in-between wereprocessed at line 152,
  3. Please check the correct orthography of the word “sympathicotonia” at line 174,
  4. Please check the space after “bars –“ at line 196.
  1. I personally would like to hear more on the meaning neural oscillations have for humans, and whether sympathetic activation in subjects with IA risks may be somewhat altered through an analysis of body position changes, since cardiovascular control settings are different in standing and lying posture. 

With Kind Regards,

Author Response

We express our gratitude to the esteemed reviewer for the work done and the detailed analysis of our article.

Point 1: Please add a space after “104.” at line 123.

Response 1: We have made these changes to the text

Point 2: Please add a space in-between wereprocessed at line 152.

Response 2: We have made these changes to the text

Point 3: Please check the correct orthography of the word “sympathicotonia” at line 174

Response 3: We have made these changes to the text

Point 4: Please check the space after “bars –“ at line 196

Response 4: We have made these changes to the text

Point 5: I personally would like to hear more on the meaning neural oscillations have for humans, and whether sympathetic activation in subjects with IA risks may be somewhat altered through an analysis of body position changes, since cardiovascular control settings are different in standing and lying posture

Response 5: We agree that the cerebral mechanisms of cardiovascular control may be different in the position of the human body lying and standing. In our study, in order to assess the dynamics of HRV indicators in the background and with HRV BF, we considered it important to use an intermediate body position in the study participants - the "sitting" position. EEG changes as neurophysiological markers in the tilt test for diagnosing syncope and epilepsy, when the patient's body position is changed, are known [Neeraj Singh et al. DOI: 10.1080/21646821.2020.1716605ÑŠ]. EEG is also used to study "Mal de Debarquement Syndrome", when bioelectrical activity is significantly dependent on body position [Yafen Chen et al. DOI:10.1109/EMBC.2018.8512699]. However, in our opinion, the above clinical conditions of a person reflect the physiological mechanisms of the deficiency of sympathetic activity, rather than real sympathicotonia. However, both passive and active orthostasis changes the movement of cerebrospinal fluid in the nervous system, which affects both the function of baroreceptors and autonomic regulation of cardiac activity, and the ratio of spectral powers of 4-7.5 Hz EEG activity in the frontal and parietal brain parts [Patent 2436503 RU, Method for detection of hemodynamic and liquorodynamic dysfunction of the brain]. That is, physical activity with a change in body position can be used to correct sympathicotonia and optimize the baroreflex in individuals with behavioral disorders, including Internet addiction. As a development of the research topic, it seems important in the future to use HRV BF technologies in various body positions to optimize both cerebral hemodynamics and brain bioelectrical activity in individuals with sympathicotonia and excessive use of Internet resources.

We have added an abbreviated version of this answer to the reviewer in the discussion section (this is highlighted in red).

Round 2

Reviewer 1 Report

Revisions made appropriately.